# Diastereo-divergent synthesis of chiral hindered ethers via a synergistic calcium(II)/gold(I) catalyzed cascade hydration/1,4-addition reaction

Xiangfeng Lin[1,2], Xia Mu [3], Hongqiang Cui[3,4], Qian Li[1,4], Zhaochi Feng[1], Yan Liu [1] ✉, Guohui Li [4] ✉ & Can Li [1] ✉

Hindered ethers are ubiquitous in natural products and bioactive molecules. However, developing an efficient method for the stereocontrolled synthesis of all stereoisomers of chiral hindered ethers is highly desirable but challenging. Here we show a strategy that utilizes in situ-generated water as a nucleophile in an asymmetric cascade reaction involving two highly reactive intermediates, 3-furyl methyl cations and *ortho*-quinone methides (*o*-QMs), to synthesize chiral hindered ethers. The Ca(II)/Au(I) synergistic catalytic system enables the control of diastereoselectivity and enantioselectivity by selecting suitable chiral phosphine ligands in this cascade hydration/1,4-addition reaction, affording all four stereoisomers of a diverse range of chiral *tetra*-aryl substituted ethers with high diastereoselectivities (up to >20/1) and enantioselectivities (up to 95% ee). This work provides an example of chiral Ca(II)/Au(I) bimetallic catalytic system controlling two stereogenic centers via a cascade reaction in a single operation.

Hindered ethers are ubiquitous in natural products and bioactive molecules, and the development of efficient synthetic methods has long fascinated organic chemists[1,2]. The traditional Williamson ether synthesis has been widely used to prepare primary alkyl ethers via $S_N2$ substitution (Fig. 1a)[3–5]. However, when secondary or tertiary alkyl halides are used as substrates, elimination side products are often obtained. Baran and co-workers reported a successful route to hindered ethers via the reaction between alcohol donors and electro-generated carbocation intermediates (Fig. 1b)[6]. However, this elegant method lacks an asymmetric version. More recently, the Fu Group developed a Cu-catalyzed enantioconvergent substitution reaction of *α*-haloamides with oxygen nucleophiles to synthesize *α*-oxygenated amides (Fig. 1c)[7]. Nevertheless, an efficient catalytic methodology for

the synthesis of chiral hindered ethers with dual chiral centers has yet to be established. Furthermore, stereo-divergent asymmetric catalysis has recently emerged as a hot research topic in organic synthesis[8–26]. This approach can afford all stereoisomers of the product, which is crucial for chemical-biological studies and the pharmaceutical industry, as different stereoisomers of chiral compounds often exhibit distinct biological activities[27–29]. Therefore, developing an efficient method for the stereocontrolled synthesis of all stereoisomers of chiral hindered ethers from readily available starting materials would be highly desirable but challenging.

Nature often utilizes water as a nucleophile in enzyme-catalyzed hydrations to synthesize chiral key structural motifs in various natural products. However, the development of artificial asymmetric hydra-

[1]State Key Laboratory of Catalysis, Dalian Institute of Chemical Physics, Chinese Academy of Sciences, Dalian 116023, PR China. [2]Key Laboratory of Bio-pesticide and Chemical Biology (Ministry of Education), College of Plant Protection, Fujian Agriculture and Forestry University, Fuzhou 350002, PR China. [3]State key Laboratory of Molecular Reaction Dynamics, Dalian Institute of Chemical Physics, Chinese Academy of Science, Dalian 116023, PR China. [4]University of Chinese Academy of Sciences, Beijing 100039, PR China. ✉e-mail: yanliu503@dicp.ac.cn; ghli@dicp.ac.cn; canli@dicp.ac.cn

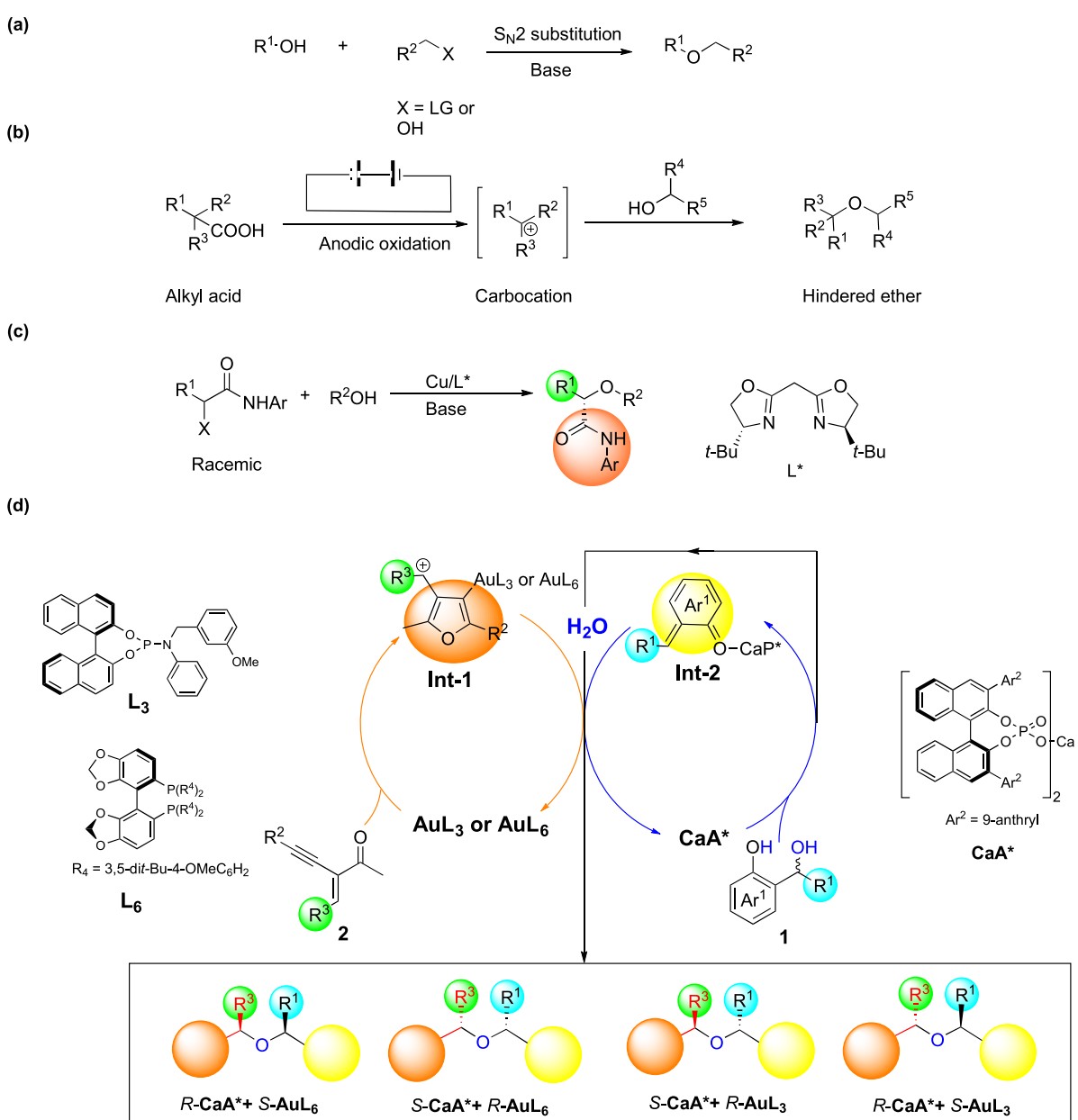

**Fig. 1 | The synthetic methods of hindered ether. a** The traditional Williamson method for the synthesis of ethers. **b** Baran's decarboxylative etherification method for the synthesis of hindered ethers. **c** Cu-catalyzed enantioconvergent substitution reaction of $a$-haloamides with oxygen nucleophiles to $a$-oxygenated amides. **d** This work: Enantio- and diastereo-divergent Ca(II)/Au(I) synergistic catalysis for synthesis of chiral hindered ether.

tions has been limited, perhaps due to the poor nucleophilicity of water[30–32]. We hypothesize that using water as the nucleophile to react with the highly reactive intermediate (HRI) could produce an alcohol that is subsequently trapped by the other HRI, resulting in the formation of a hindered ether. *Ortho*-quinone methide (*o*-QM) immediately came to mind as suitable HRI because they can be generated and activated by chiral phosphoric acid (CPA) or Lewis acid from *o*-hydroxybenzyl alcohols through dehydration[33–51]. The in situ released water may serve as a nucleophile to initiate the subsequent cascade reaction. Additionally, we selected 3-furyl methyl cation as the other HRI because this specie can be generated by Au(I) complex-catalyzed intramolecular cyclization from 2-(1-alkynyl)−2-alken-1-one[52–61]. Thus, an asymmetric version of this cascade reaction can be achieved through a highly efficient chiral acid/Au(I) complex synergistic catalytic system by electrophilic activation of *o*-QMs with chiral acid and activation of 3-furyl methyl cations with chiral Au(I) complex[62–69].

We present herein a bimetallic catalytic system consisting of chiral Ca(II) and Au(I) catalysts, which facilitates a cascade reaction between in situ generated water and both 3-furyl methyl cations and *o*-QMs. This transformation affords a broad range of chiral tetra-aryl substituted ethers in moderate to high yields, with impressive levels of diastereoselectivities (up to >20/1) and enantioselectivities (up to 95% ee) (Fig. 1d). Specifically, we have discovered that chiral BINOL-derived calcium phosphate catalysts[70–80] serve as efficient Lewis acids for the generation and activation of *o*-QMs. By carefully selecting a suitable chiral Au(I) complex in combination with the chiral calcium phosphate catalyst, we have achieved stereodivergent cascade reactions, providing access to all four stereoisomers of the products (Fig. 1d).

## Results and discussion
Initially, we selected 2-(1-alkynyl)−2-alken-1-one **2a** as one of the model substrates and (R)-**L₁**[(NCMe)AuSbF₆] as the catalyst to generate

3-furyl methyl cations. To generate o-QMs, we chose 2-(hydroxylmethyl) phenol **1a** and employed chiral phosphoric acids as cocatalysts in this reaction. The cascade reaction afforded the *anti*-hindered ether **3a** as the major product in good yield (83%) in DCE at −25 °C (Table 1, entry 1). However, the product's ee was only 5%. We attempted to identify the optimal catalyst by evaluating 3,3′-disubstituted BINOL-based chiral phosphoric acid but couldn't improve the ee value of **3a** (see the Supplementary Table 1). Nonetheless, we discovered that a chiral phosphoric acid (**H[A]**) directly purified on silica gel without washing with aqueous HCl could offer good enantioselectivity (80% ee) in an accidental experiment (Table 1, entry 2). This serendipitous finding motivated us to investigate why the same chiral phosphoric acids with different purification processes resulted in distinct stereoinduction. Ishihara's group previously found that the chiral phosphoric acid purified on silica gel without washing with HCl could function as a Lewis acid to catalyze a highly enantioselective direct Mannich-type reaction in 2010[71]. Based on FAB-LRMS analysis, they suggested that this **H[A]***was composed of calcium phosphate and sodium phosphate. Inspired by this work, we examined the cascade reaction of in situ generated o-QMs and 3-furyl methyl cations by using a series of alkali or alkaline-earth metal phosphates as cocatalysts. As predicted, Li[I], Na[I], and Mg[II] salts showed disappointing results (Table 1, entries 3-5). However, the Ca[II] salt ((S)-**Ca[A]₂**) and Sr[II] salt ((S)-**Sr[A]₂**) could smoothly promote the cascade reaction, yielding the hindered ether **3a** in moderate yield with 79% and 60% ee, respectively (Table 1, entries 6 and 7). Our efforts to identify the optimal catalyst by evaluating 3,3′-disubstituted BINOL-based calcium phosphates proved that 9-anthryl substituted calcium phosphate (S)-**Ca[A]₂** was the best catalyst (see the Supplementary Table 1).

Subsequently, several Au(I) complexes were evaluated, however, only (NCMe)Au(I)SbF₆ delivered the desired product **3a** in good yield (see the Supplementary Table 1). The main optimization efforts focused on the chiral ligands in Au(I) complexes. Initially, a series of chiral mono phosphoramidites with different amido substituents were screened, and (R)-**L₃**[(NCMe)AuSbF₆] provided **3a** with the highest ee values (88% ee) (Table 1, entries 6 and 8 vs entry 9). Further investigations suggested that *R*-configured phosphoramidites resulted in slightly higher diastereoselectivities and enantioselectivities than their antipode (Table 1, entry 9 vs entry 10). Solvent screening showed that DCE yielded the best yield and selectivity (Table 1, entry 9 vs entries 11 and 12). Notably, a reduction in the catalyst loading of Au(I) complex to 5 mol% increased the ee value of the product **3a** to 90% (Table 1, entry 13). Consequently, the optimized conditions involved the use of (S)-**Ca[A]₂** and (R)-**L₃**[(NCMe)AuSbF₆] as catalysts in DCE at −25 °C (Table 1, entry 13).

Surprisingly, the *syn*-3-furyl substituted ether **3a** ratio was significantly increased to 1:1 from 1:10 when bisphosphine ligands (R)-**L₄** and (R)-**L₅** were evaluated (Table 1, entries 14 and 15) in this cascade reaction. Encouraged by these results, we further evaluated hindered bisphosphine ligand (R)-**L₆**. To our delight, with (R)-**L₆** as the bisphosphine ligand, the reaction yielded mainly the *syn*-diastereoisomer **4j** with 4:1 d.r. and 93% ee (Table 1, entry 16). (S)-**L₆**[(NCMe)AuSbF₆]₂ was also used instead of (R)-**L₆**[(NCMe)AuSbF₆]₂ as the Au(I) catalyst. However, different from **L₃**[(NCMe)AuSbF₆] as the catalyst, (S)-**L₆**[(NCMe)AuSbF₆]₂ afforded the other enantiomer of **4j** with a 1:4 d.r. and 93% ee (Table 1, entry 17), suggesting that the enantioselectivity of the reaction might be controlled by the Au(I) complex.

Under the optimized reaction conditions, we explored the substrate scope with various o-QM precursors to investigate the generality of this asymmetric cascade reaction with 2-(1-alkynyl)-2-alken-1-one **2a** as the other substrate (Fig. 2). Pleasingly, the reactions were highly compatible with either electron-donating or electron-withdrawing substituents at the benzene ring of o-QMs (**1a-l**), delivering the corresponding products in consistently good yields with excellent diastereo- and enantioselectivities. However, *ortho*-substituted aryl

groups had an detrimental effect on the ee value of this reaction, likely due to steric hindrance effects (**3f**). Additionally, o-QMs with 4-biphenyl, 2-naphthyl, and 1-naphthyl substituents were also well accommodated, affording the desired products (**3j-l**) with good yields and high enantioselectivities (91%, 90%, and 78% ee, respectively). Substituents at the quinone methide fragment were also tolerated, and the corresponding products **3m-n** were obtained in good to excellent enantioselectivities (86-90% ee). Notably, even alkyl-substituted o-QM was shown to be suitable acceptors for the reaction, giving rise to the corresponding product **3o** with 74% ee. When (R)-**L₆** was changed to (R)-**L₃**, a series of *syn*-diastereoisomers were obtained in highly enantiomerically enriched form (**4a-h**) with excellent ee values ranging from 88% to 95%. These results demonstrate that the current method provides a reliable and powerful protocol for stereodivergent access to optically hindered ethers. Alkyl-substituted o-QM was proved unavailable in this catalytic system.

After exploration of the reaction scope of o-QM precursors, the effects of 2-(1-alkynyl)-2-alken-1-ones were subsequently evaluated. Fig. 3 demonstrates that 2-(1-alkynyl)-2-alken-1-ones, which vary in electronic characteristics and substitution patterns on the benzene ring, are all compatible with the reaction conditions. This broad tolerance leads to the production of the corresponding *anti* products **3p-w** in satisfactory yields, accompanied by superior diastereo- and enantioselectivities, with enantiomeric excesses as high as 93%. In general, electron-donating groups in two aryl groups of 2-(1-alkynyl)-2-alken-1-ones furnished the products in higher yields and ee values than those with electron-withdrawing substituents. In particular, with Johnphos(NCMe)AuSbF₆ as Au(I) complex instead of (R)-**L₃**[(NCMe)AuSbF₆], electron-withdrawing chiral hindered ether **3w** was isolated in 40% yield and 67% ee. However, alkyl-substituted 2-(1-alkynyl)-2-alken-1-ones were prove to be unavailable in this catalytic system. Similarly, *syn*-diastereoisomers (**4j-m**) could be accessed by a combination of catalysts (R)-**L₆**[(NCMe)AuSbF₆]₂ and (S)-**Ca[A]₂** (73-93% ee). In addition, 2-(1-alkynyl)-2-alken-1-one with alkyl substituted at $R^2$ position was well tolerated for the reaction, giving rise to the corresponding product **4n** with 86% ee. 2-(1-alkynyl)-2-alken-1-one with alkyl substituted at $R^3$ position could not work in this catalytic system.

To further demonstrate the practicality of this reaction, the cascade reaction of **1b** and **2a** was carried out on a gram scale under optimized reaction conditions. The corresponding product, the hindered ether **4a**, could be obtained with 52% yield, 4:1 d.r. and 90% ee. **3b** was obtained with 42% yield, 8:1 d.r. and 90% ee (Fig. 4a). To assess the synthetic utility of this methodology, the predominant diastereomer of compound **4j** was subjected to a ring-opening reaction with m-CPBA, delivering the compound **5** with a yield of 64%. Furthermore, protection of the hydroxy groups on **5** with p-bromobenzoic acid readily delivered compound **6a** in 55% yield and nearly without loss of enantiopurity (Fig. 4b). Moreover, tribromo substituted product **6b** was prepared following the same procedures in order to determine the configuration of product. The absolute configurations of **3** and **4** were determined to be (R, R) and (R, S) by X-ray crystallography of **6b** coupled with the results of CD experimental and theory computational spectra[81–83] (see the Supplementary Fig. 2). Additionally, the hydroxyl group of the major diastereomer of **4j** was selectively shielded using propargyl. Subsequently, a Cu-catalyzed cycloaddition was employed to synthesize **7**, incorporating a triazole skeleton, achieving a commendable 76% yield and 93% ee through the application of click chemistry. In addition, **4a** can be readily triflated to facilitate subsequent efficient cross-coupling, yielding biaryl **8**, while preserving the benzylic stereocenter. Importantly, these transformations exhibit no discernible degradation in enantiopurity.

To demonstrate the potential for enantio- and diastereodivergent synthesis of the entire set of stereoisomeric products, we conducted on a series of cascade experiments under the optimized reaction conditions. These experiments involved the strategic combination of

**Table 1 | The optimization of the reaction**

| Entry | (S)-Phosphate | L | Solvent | Yield/%[a] | Ee/% (major)[b] | D.r. (anti/syn)[c] |
|---|---|---|---|---|---|---|
| 1 | H[A] | (R)-L$_1$ | DCE | 83 | 5 | 10/1 |
| 2[d] | H[A]* | (R)-L$_1$ | DCE | 62 | 80 | 10/1 |
| 3 | Li[A] | (R)-L$_1$ | DCE | 0 | N.D. | N.D. |
| 4 | Na[A] | (R)-L$_1$ | DCE | 0 | N.D. | N.D. |
| 5 | Mg[A]$_2$ | (R)-L$_1$ | DCE | 0 | N.D. | N.D. |
| 6 | Ca[A]$_2$ | (R)-L$_1$ | DCE | 72 | 79 | 10/1 |
| 7 | Sr[A]$_2$ | (R)-L$_1$ | DCE | 66 | 60 | 10/1 |
| 8 | Ca[A]$_2$ | (R)-L$_2$ | DCE | 65 | 81 | 15/1 |
| 9 | Ca[A]$_2$ | (R)-L$_3$ | DCE | 68 | 88 | 10/1 |
| 10 | Ca[A]$_2$ | (S)-L$_3$ | DCE | 66 | 85 | 10/1 |
| 11 | Ca[A]$_2$ | (R)-L$_3$ | DCM | 61 | 77 | 10/1 |
| 12 | Ca[A]$_2$ | (R)-L$_3$ | CHCl$_3$ | 30 | 75 | 8/1 |
| 13[e] | Ca[A]$_2$ | (R)-L$_3$ | DCE | 67 | 90 | 10/1 |
| 14[f] | Ca[A]$_2$ | (R)-L$_4$ | DCE | 40 | N.D. | 1/1 |
| 15[f] | Ca[A]$_2$ | (R)-L$_5$ | DCE | 64 | N.D. | 1/1 |
| 16[f] | Ca[A]$_2$ | (R)-L$_6$ | DCE | 65 | 93 | 1/4 |
| 17 | Ca[A]$_2$ | (S)-L$_6$ | DCE | 54 | −93 | 1/4 |

All reactions were carried out on a 0.1 mmol scale with 1 eq precursor of o-QMs **1a**, 1.05 eq **2a**, 10 mol% of phosphate, and 10 mol% of **L**[(NCMe)AuSbF$_6$] in DCE (1 mL) at −25 °C.

N.D. no detection, DCE dichloroethane, DCM dichloromethane.

[a] Isolated yield.
[b] Determined by chiral HPLC.
[c] Determined by crude 1H-NMR.
[d] **H[A]*** was purified on silican gel without washing with HCl.
[e] 5 mol% of **L**[(NCMe)AuSbF6].
[f] 2.5 mol% of **L**[(NCMe)AuSbF$_6$]$_2$.

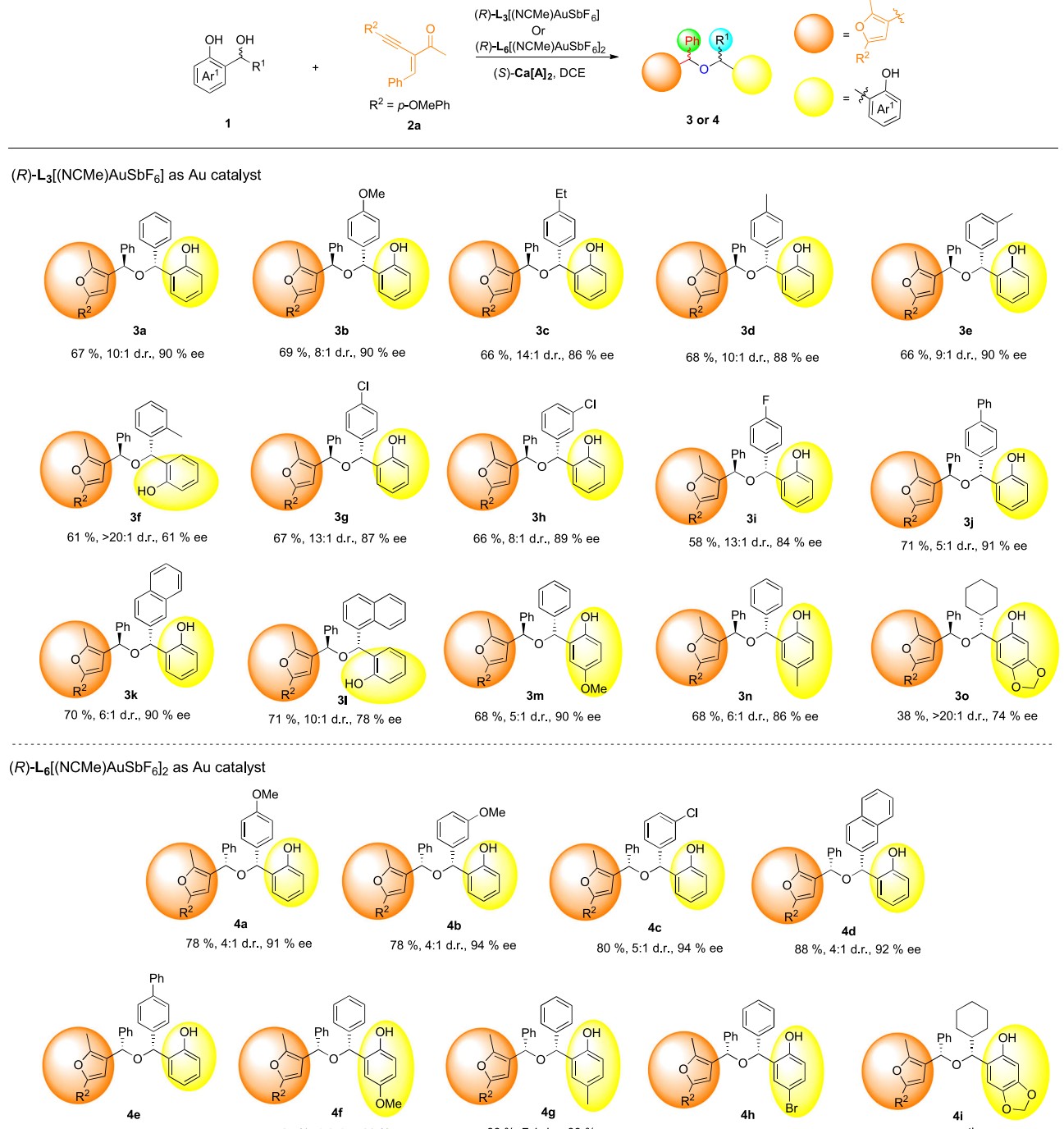

**Fig. 2 | Substrates scope of *o*-QMs.** All reactions were carried out on a 0.15 mmol scale with 1 eq precursor of *o*-QMs **1**, 1.05 eq **2a**, 10 mol% of (*S*)-**Ca[A]₂** and 5 mol% of (*R*)-**L₃**[(NCMe)AuSbF₆] or 2.5 mol% of (*R*)-**L₆**[(NCMe)AuSbF₆]₂ in DCE (1 mL) at −25 °C. Isolated yield. d.r. was determined by crude ¹H-NMR and ee values were determined by chiral HPLC.

the appropriate enantiomer of the chiral calcium phosphate **Ca[A]₂** with two different gold catalysts, **L₃**[(NCMe)AuSbF₆] and **L₆**[(NCMe)AuSbF₆]₂. Remarkably, by utilizing the four available catalyst combinations, a stereodivergent synthesis of the complete matrix of four stereoisomeric hindered ethers **3b**, **3b'**, **4a**, and **4a'** was achieved. This approach enabled us to access these stereoisomers in a highly diastereo- and enantioselective fashion, showcasing the versatility and efficiency of the catalytic system (Fig. 5).

In order to confirm the mechanism in this cascade reaction, we investigated the cascade reaction between *o*-QMs precursor **1a** and 2-(1-alkynyl)-2-alken-1-one **2b** with 1 equiv H₂O¹⁸ as additive (Fig. 6a). O¹⁸ marked hindered ether **9** was detected by FTMS (see the Supplementary Fig. 1), which suggests that water generated from *o*-QM precursor may act as a key reaction intermediate in this cascade reaction. When the reaction involving (±)-**1a** was run to partial conversion, **1a** was recovered without enantioenrichment, whereas **3p** or **4i** was obtained in 88% or 73% ee (Fig. 6b, c). This data evokes the kinetic resolution is not operative. Moreover, we synthesis protected *o*-hydroxybenzyl alcohol **10** for this controlled experiment. No product was obtained in the reaction between **10** and **2b**, which means *o*-QM is the reaction intermediate. (Fig. 6d).

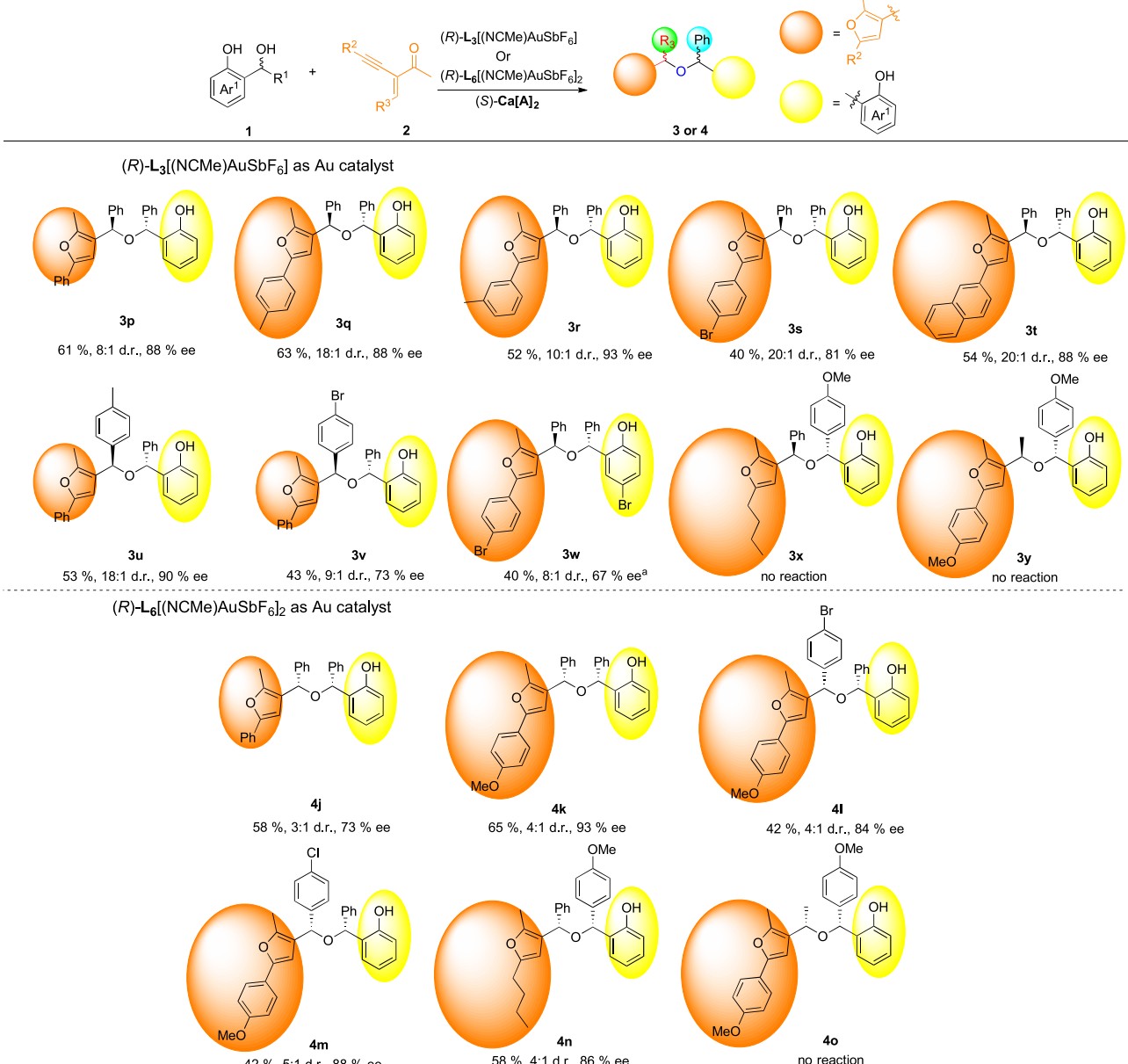

**Fig. 3 | Substrates scope of 2-(1-alkynyl)-2-alken-1-ones and _o_-QMs.** All reactions were carried out on a 0.15 mmol scale with 1 eq precursor of _o_-QMs **1**, 1.05 eq **2**, 10 mol% of (_S_)-**Ca[A]₂** and 5 mol% of (_R_)-**L₃**[(NCMe)AuSbF₆] or 2.5 mol% (_R_)-**L₆**[(NCMe)AuSbF₆]₂ in DCE (1 mL) at −25 °C. Isolated yield. d.r. was determined by crude ¹H-NMR and ee values were determined by chiral HPLC. ᵃJohnphos(NCMe)AuSbF₆ served as Au complex instead of (_R_)-**L₃**[(NCMe)AuSbF₆].

To probe into the mechanism of cascade reaction, we carried out DFT calculations by using **1a** and **2b** as the substrates. According to the earlier computation results[78], calcium is coordinated with two chiral phosphates. Considering the propensity of calcium ions, we have created two distinct calcium coordination models (Fig. 7). In the model A, the calcium ion is hexacoordinated, and the ligands are three methanol molecules from the synthetic solvent, two oxygen atoms on phosphoric acid (each phosphoric acid molecule contributes one oxygen atom), and one water molecule. This model is similar with the transition state found by theoretical in ref. 78. The Ca ion still has the same coordination numbers in the B model, but two oxygen atoms are used in place of two methanol molecules. This implies that the metal receives two oxygen atoms from each phosphoric group. For studying the selectivity by chiral Ca phosphate, we selected two models. They are _S_-**AuL₃** + _S_-**CaA\*** and _R_-**AuL₆** + _S_-**CaA\***. The whole catalysis reaction can be divided into three stages: the first only involves the reaction of the alkyne with **AuL₃**, the second involves the attack of water molecule

on the byproduct of the frontier reaction with the help of **CaA\***, and the final stage involves a nucleophilic addition with the help of phosphate. The 9-anthryl group on the **CaA\*** molecule is full maintained in **L₃** system and frozen after several optimizations. However, in **L₆** system, this group is reduced to a benzene ring only in the final stage that is also frozen through several optimizations.

We used the theory study for the _S_-**AuL₃** + _S_-**CaA\*** and _R_-**AuL₆** + _S_-**CaA\*** reaction process and identified the rate-limiting step for each system to explain the enantioselectivity results from experiment. All calculations were carried out with the Gaussian16 program package[84]. Molecular geometries were optimized with the PBE0 functional[85,86]. The 6-31 G\* basis set was used for the C, H, O, N, and P atoms, and the SDD effective core potential (ECP)[87–90] for Ca and Au. Considering the large system size and many aromatic rings, we added a long-range D3 version of Grimme's dispersion with Becke-Johnson damping[91] for all computations. Frequency calculation at the same level were performed to characterize the stationary points as minima or transition

**Fig. 4 | Practicality of the reactions. a** Scale-up reaction. **b**, **c** synthetic transformation.

state. In general, intrinsic reaction coordinate[92] calculation would give reactants and products corresponding to transition state. Single-point energies were estimated by PBE0 with def2TZVP[93,94] for all atoms under the PCM solvation model[95] with dichloroethane. The energies given throughout the paper are Gibbs free energy computed at 298 K in kcal/mol.

The mechanism and energies of the *S*-**AuL₃** + *S*-**CaA\*** reaction is shown in Fig. 7. All the stationary state throughout the reaction is showed, and corresponding geometries depicted in SI. The catalytic cycle begins with the interactions between the gold-complex catalyst and the triple bond on substrate **2b**. From the transition state *S*-**L₃**-TS1, a nucleophilic attack at the oxygen and carbon neighbor the **AuL₃** bonding site required a 3.3 kcal/mol activation energy and

resulted in the positive charged five-member ring products *S*-**L₃**-INT1. This intermediate complex has a lower energy than reactant of about -16.7 kcal/mol, and this intermediate serves as the starting point for the enantioselectivity since a big molecule **CaA\*** is participated in next reaction. As described before, the *S*-**CaA\*** has two models, thus Fig. 7 depicts two substrates with two configuration products and four transition states. Only the Au complex at a mirror state in four transitions, which results in an *R* or *S* product. With different **CaA\*** models and TS configurations, the energy barrier varies. The *S*-**L₃**-TS2B-*R* has the lowest energy barrier at 10.2 kcal/mol. Based on the **L₃**-TS2 structure, a certain phosphate group is necessary for water activation, resulting a closer distance between 9-anthryl on **CaA\*** and **L₃**-INT1. When **L₃**-TS2 is *R*-type, the five-member ring and its methyl

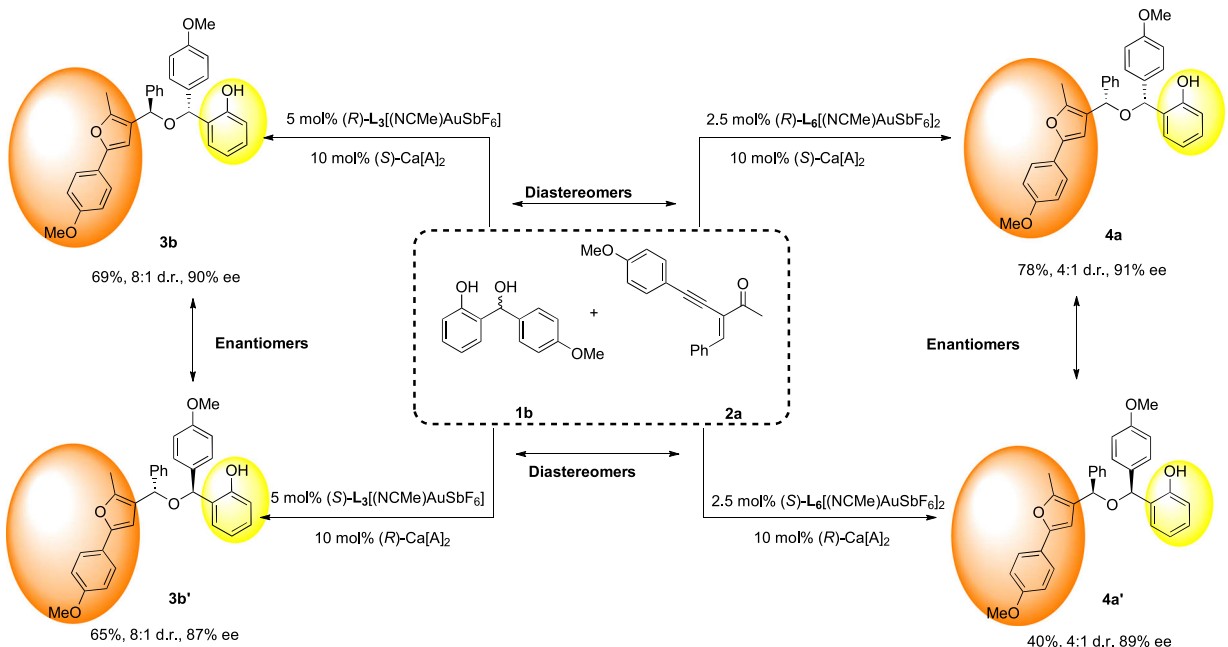

**Fig. 5 | Divergent synthesis of all four isomers of chiral hindered ether** 3b **or** 4a. All reactions were carried out on a 0.15 mmol scale with 1 eq precursor of 1b, 1.05 eq 2a, 10 mol% of (*S*)-Ca[A]₂ and 5 mol% of (*R*)-L₃[(NCMe)AuSbF₆] or 2.5 mol% of (*R*)- L₆[(NCMe)AuSbF₆]₂ in DCE (1 mL) at −25 °C. Isolated yield. d.r. was determined by crude ¹H-NMR and ee values were determined by chiral HPLC.

**Fig. 6 | Controlled experiment. a** The reaction between (±)-1a and 2b with 1 equiv H₂O¹⁸ as additive. **b** The chiral calcium phosphate **Ca[A]₂** and **L₃**[(NCMe)AuSbF₆] catalyzed reaction between (±)-1a and 2b was run to partial conversion. **c** The chiral calcium phosphate **Ca[A]₂** and **L₆**[(NCMe)AuSbF₆]₂ catalyzed reaction between (±)-1a and 2b was run to partial conversion. **d** The reactions between protected *o*-hydroxybenzyl alcohol **10** and 2b under standard conditions.

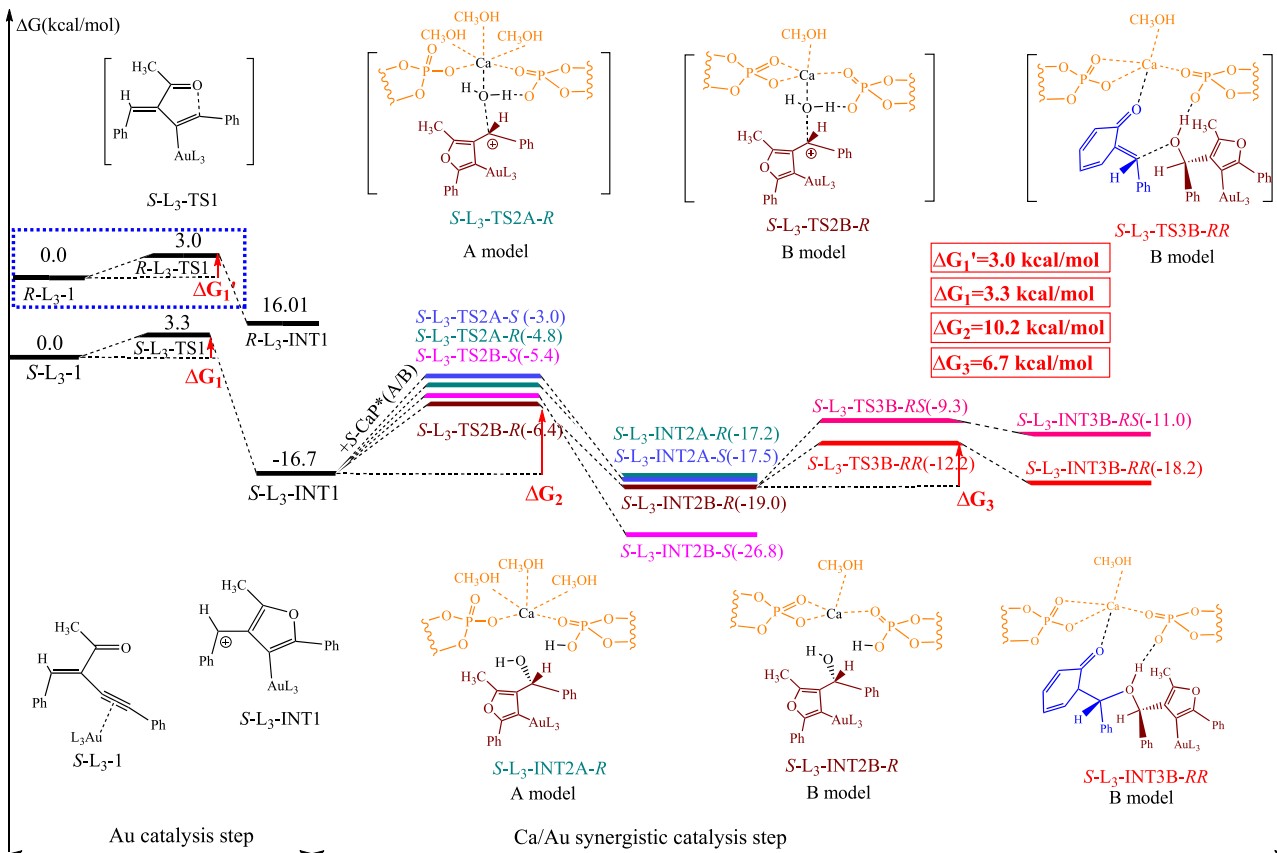

**Fig. 7 | The DFT studies of *S*-AuL₃ + *S*-CaA\* model.** This figure depicts reaction paths in (*R*)-L₃[(NCMe)AuSbF₆] catalytic system. TS1 means transition state 1. INT1 means intermediate 1. TS2A-*S*, TS2A-*R*, TS2B-*S* and TS2B-*R* means transition state 2 in two model with different absolute configurations. INT2A-*R*, INT2A-*S*, INT2B-*R* and INT2B-*S* means intermediate 2 in two model with different absolute configurations. TS3B-*RS* and TS3B-*RR* means transition state 3 in B model with different absolute configurations. INT3B-*RS* and INT3B-*RR* means intermediate 3 in B model with different absolute configurations.

group of **AuL₃** can stay away from the 9-anthryl group. When **L₃**-TS2 is in the *S* configuration, the distance is much closer. Thus, **L₃**-TS2-*R* is favorable. In addition, we founded that the Ca prefers to maintain two coordination with phosphatic, as evidenced by the fact that each of the two transitions from the B model-**CaA\*** have a lower activation energy than the A model-**CaA\***. Along the reaction, the energy barrier is 6.7 kcal/mol for **L₃**-TS3B-*RR* and 9.7 kcal/mol for **L₃**-TS3B-*RS*, respectively. Combining the **L₃**-TS3 structure, a π-π stacking between five-member ring and benzene on substrate **1a** is observed in *R*-type, but not in *S*-type, which means a longer reaction distance between C-OH. This bond is shortened from 1.93 Å in the **L₃**-TS3-*R* to 1.76 Å in **L₃**-TS3-*S*, resulting a *R*-favorable configuration product. Besides, we also calculated the first step of *R*-**AuL₃** react with substrate **2b**, only 3.0 kcal/mol activation is obtained. In summary, the barrier for the first stage involving AuL₃ is merely 3.3 kcal/mol, whereas barrier for the Ca/Au synergistic catalysis step is at least 6.7 kcal/mol. We concluded that the Ca/Au synergistic catalysis is the rate-limiting step in catalytic process.

The mechanism and energies of the reaction between *R*-**AuL₆** + *S*-**CaA\*** is shown in Fig. 8. Unlike in **AuL₃** system, the activation energy of the Au catalysis step in **AuL₆** system increased to 8.4 kcal/mol, a value that is significantly higher than that of **L₃** system. This finding suggests that the larger **L₆** molecule's steric effect will decrease the reaction activity of Au and triple bonds. The subsequent nucleophilic addition of charged products and water molecules similarly has four transition states (*R*-**L₆**-TS2A-*R*, *R*-**L₆**-TS2B-*R*, *R*-**L₆**-TS2B-*S* and *R*-**L₆**-TS2A-*S*). The energies revealed that the **L₆** system was more likely to yield *S*-product, whereas the A model, which had a relatively loose Ca spatial coordination and a lower activation energy, became the primary reaction

path. *R*-**L₆**-TS2A-*S* has the lowest activation among the four transition states, which is roughly 5.2 kcal/mol. Similarly, *R*-**L₆**-INT2A-*S* was the lowest energy intermediate. According the **L₆**-TS2 structure, a larger distance between the **L₆**-INT1 and **CaA\*** than **L₃**-TS2 is founded due to the steric hindrance of the large group **L₆**. As a result, the repulsion between the 9-anthryl group on phosphoric acid which is not participated in water activation and **L₆**-INT1 becomes the major interaction for chiral selectivity. *S*-type is the favorable configuration. At the final stage, the -OH group on the INT2 would assault the double bond by aiding with a phosphatic oxygen. The energy barriers for the INT2A-*S* complexes are 6.3 kcal/mol for *R*-**L₆**-TS3A-*SR* and 15.7 kcal/mol for *R*-**L₆**-TS3A-*SS*. *RR*-type is energetic favorable. Unlike **L₃**-TS3 system, the π-π stacking is not observed in **L₆**-TS3 structure. But a same loose transition structure for *R*-configuration is located. The C-OH distance changes from 1.89 Å in **L₆**-TS3A-*SR* to 1.74 Å in **L₆**-TS3A-*SS*. We supposed that the selectivity of **L₆**-TS3 may be a comprehensive result from the **L₆** group. Similar calculations were made for the reaction involving *S*-Au**L₆** and substrate **2b**, and an energy barrier about 8.4 kcal/mol was found. The DFT studies suggested that the initial Au catalysis step has the highest activation energy and is the rate-limiting step, which is consistent with experimental data.

In summary, we developed an asymmetric cascade reaction of in situ generated H₂O with 3-furyl methyl cations and *o*-QMs catalyzed by a highly efficient chiral BINOL-derived calcium phosphate/chiral Au(I) complex bimetallic catalytic system. Importantly, these two chiral catalysts allow for full control over the configuration of the stereocenters, affording all four stereoisomers of a diversity of chiral *tetra*-aryl substituted ethers in moderate to high yields and with high levels of diastereoselectivities (up to > 20/1) and enantioselectivities (up to

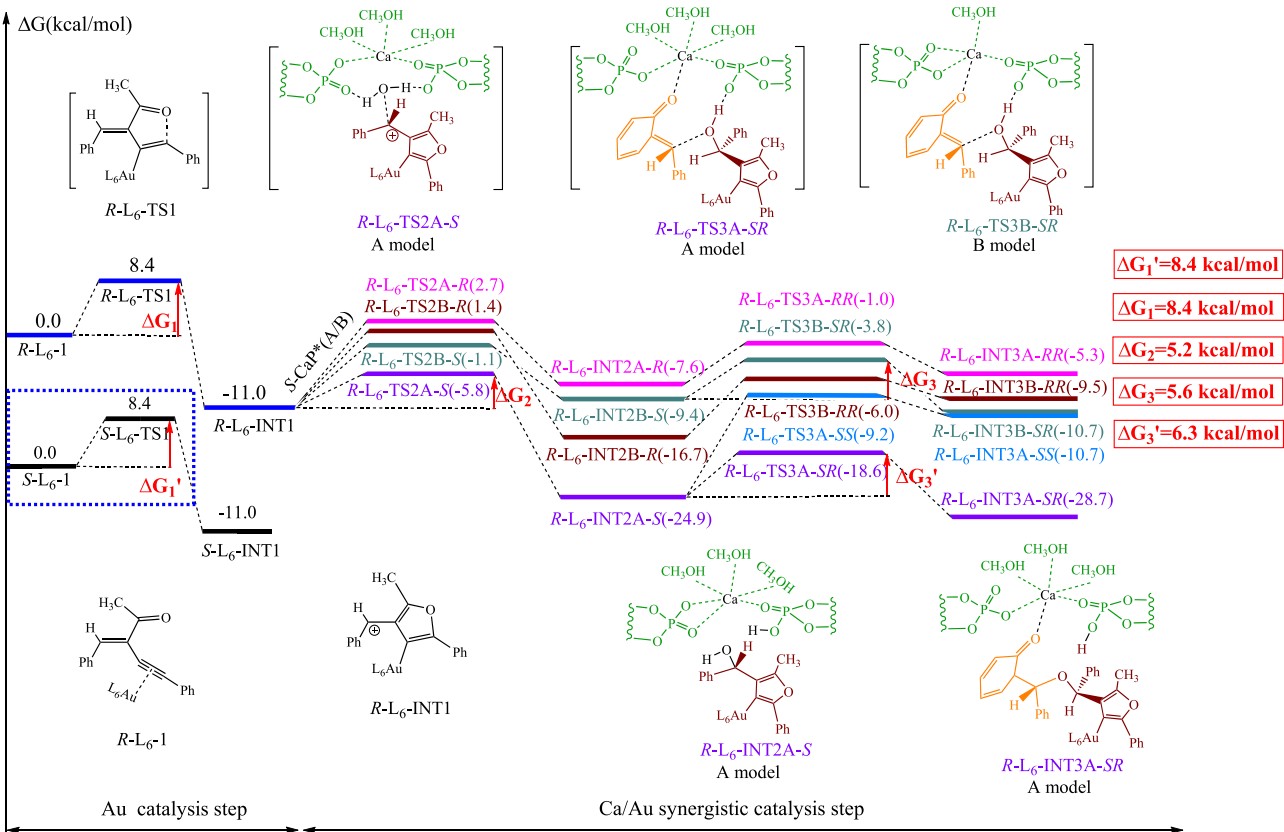

**Fig. 8 | The DFT studies of *R*-AuL₆ + *S*-CaA\* model.** This figure depicts reaction paths in (*R*)-**L₆**[(NCMe)AuSbF₆]₂ catalytic system. TS1 means transition state 1. INT1 means intermediate 1. TS2A-*S*, TS2A-*R*, TS2B-*S* and TS2B-*R* means transition state 2 in two model with different absolute configurations. INT2A-*R*, INT2A-*S*, INT2B-*R* and INT2B-*S* means intermediate 2 in two model with different absolute configurations. TS3A-*RR*, TS3B-*SR*, TS3B-*RR*, TS3A-*SS* and TS3A-*SR* means transition state 3 in two model with different absolute configurations. INT3A-*RR*, INT3B-*SR*, INT3B-*RR*, INT3A-*SS* and INT3A-*SR* means intermediate 3 in two model with different absolute configurations.

95% ee). The mechanism studies indicated that H₂O generated from *o*-QM precursor is a key reaction intermediate, and calcium phosphate acts as a shuttle, absorbing and activating the in situ-generated H₂O, which then attacks the 3-furyl methyl Au(I) complex. The current work not only develops an asymmetric catalytic reaction for the synthesis all stereoisomers of hindered ethers but also provides a rare example of chiral Ca(II)/Au(I) bimetallic catalytic system controlling two stereogenic centers via a cascade reaction in a single operation.

## Methods

### General experimental procedure of asymmetric cascade reaction

To a 10-mL test tube were sequentially added (*R*)-**L₃**[(NCMe)AuSbF₆] (0.0075 mmol, 7.5 mg) or (*R*)-**L₆**[(NCMe)AuSbF₆]₂ (0.00375 mmol, 8.0 mg), Ca[A]₂ (0.015 mmol, 21.6 mg) and DCE (2.0 mL). Substrate **2** (0.16 mmol, 1.05 eq) and *o*-QM precursor **1** (0.15 mmol) were added in turn to the solution at −25 °C. The reaction mixture was monitored by TLC. Upon completion, the residual was purified by silica gel flash chromatography (petroleum ether: ethyl acetate, 20: 1) to afford the desired product **3** or **4**. The racemic examples were prepared by the catalysis of JohnphosAu(NCMe)SbF₆ and Sc(OTf)₃ in room temperature.

## Data availability

Crystallographic data has been deposited in the Cambridge Crystallographic Data. Center under accession number CCDC: 2125710. These data can be obtained free of charge from The Cambridge Crystallographic Data Centre via https://www.ccdc.cam.ac.uk/structures/Search?access=referee&ccdc=2125710&Author=Xiangfeng+Lin+xflin.

Source data are present. All data are available from the corresponding author upon request. Source data are provided with this paper.

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

## Acknowledgements

This work was financially supported by the National Natural Science Foundation of China (22271276, 21871254, 22288101) and National Key Research and Development Program of China (No. 2022YFC2105900).

## Author contributions

X.L. performed the experiments and wrote the article. X.M. performed the theory calculation of mechanism. H.C. performed the theory calculation of CD theory computational spectra. Q.L., Z.F. and G.L. participated in some discussions. Y.L. and C.L. conceived the concept and supervised the research project.

## Competing interests

The authors declare no competing interests.
