## [Peer Review File · Nature Communications]

Diastereo-divergent Synthesis of Chiral Hindered Ethers via a Synergistic Calcium(II)/Gold(I) Catalyzed Cascade Hydration/1,4-Addition ReactionREVIEWER COMMENTS

Reviewer #1 (Remarks to the Author):

This work realizes a relay and enantioselective reaction using combined chiral gold and chiral calcium phosphate catalysts. The relay catalysis is sure to be unprecedented although the key intermediates in the first step of gold catalysis is well established. The second step using a 1:1 reactant ratio, showing a dynamical kinetic resolution. Furthermore, there are DFT calculations to show the energy profile of the proposed reaction pathway. Overall speaking, this work is novel because Ca(+2) is very crucial as the acid (H+1) and Na(+1) gave poor enantioselectivity or poor diastereoselectivity.

A combined relay catalysis to yield furanyl ethers through a carbocation intermediate is also challenging although some reports of chiral phosphate are known to be effective albeit with limited cases. This reviewer is enthusiastic to recommend this work in this journal after minor revision.

(1) In Figure 1, the proposed catalytic cycle shows the ligands L* and A* but the left hand has the ligands L3 and L6; these ligands are actually the same and should be clearly labelled by L3 and L6 in the catalytic cycle.

(2) In Figure 2, entries 9 and 10 are testing two enantiomers of L3 ligands; but the outcome of ee are nearly the same. But in entries 16 and 17, the resulting enantiomers are reverse. Why there is so different for the outcome of enantioselectivity.

(3) In Figure 2, N.D is not clear to me. Is no reaction taking place or no enantioselectivity or no diastereoselectivity.

(4) In the Data Availability, there are x-ray crystallographic work to be deposited at Cambridge Crystallographic Data center. For the convenience of readers; ORTEP images of 3a and 4j (line 182) must provided in the text as I almost reject this paper when I did not see these very important data in the text.

(5) There two x-ray structures mentioned in the text, but only CCDC 2125710 is mentioned line 319?

(6) DFT Figures in Figure 8 and 9 are very small to show the clarity. I am unable to make a suggestion and probably a vertical presentation is better.

Reviewer #2 (Remarks to the Author):

A Ca-/Au-cocatalyzed, enantio- and diastereoselective reaction of α -alkynyl- α,β -unsaturated ketones with ortho-hydroxy benzyl alcohols via transient ortho-quinone methides, 3-furyl methyl cations, and in situ generated water is reported giving rise to hindered chiral tetra-aryl substituted ethers. Moderate to good yields and good to very good levels of diastereoselectivities and enantioselectivities were reported. Depending upon the proper choice of catalyst combination all four enantio- and diastereomers were available at will. Mechanistically, the water generated in situ from dehydration of the ortho-hydroxy benzyl alcohol attacks both the 3-furyl methyl Au(I) complex and the ortho-quinone methide. Altogether, a nice combination of chiral Ca(II)/Au(I) bimetallic catalysis controlling two stereogenic centers via a cascade reaction in a single operation. DFT-calculations corroborate the mechanistic picture outlined above.

While I find the work has been carried with great care and reports excellent selectivities, dual or cooperative catalysis has been established quite a number of times with organo/metal phosphate catalysis limiting the originality of the work. In addition, the products are rather

special in their structure and I am wondering what they can be employed for. The ensuing synthetic transformation reported in Figure 5 does not convince the reader of the synthetic value of this transformation.

Reviewer #3 (Remarks to the Author):

Chiral hindered ethers are ubiquitous motifs in natural and bioactive products. In this manuscript, Li and co-workers reported a method of synthesizing chiral hindered ethers by Calcium(II)/Gold(I) synergistic catalysis. The combined catalyst system features the use of chiral Ca(II) phosphate for the generation and activation of o-QMs, while chiral Au(I) catalyst is used for the generating and activating 3-furyl methyl cations. These two chiral catalysts allow for full control over the configuration of the stereocenters, and four stereoisomers of the tetra-aryl substituted ethers can be obtained by selecting suitable chiral ligands. Despite these advantages, however, the completeness and richness of this work may not be sufficient for publication in Nature Communications.

Comments:

1. The structure of the product is inconsistent with that of the reactants in Fig 1a, and in Fig 1d, substrate 1 generates Int-2 under the action of Catalyst CaA*, hence CaA* should be placed on the right side of Fig 1d, while Ligands L3 and L6 should be positioned on the left side for the clarity.
2. In Fig 6, the authors mentioned compounds 3b' and 4a', however, their detailed information was not found in the SI.
3. The authors propose that under Au(I) catalysis, 2-(1-alkynyl)-2-alken-1-ones undergo transformation into a positively charged five-membered ring, subsequently forming alcohols when reacting with water. The isolation of the alcohols may be tried to further substantiate this catalytic mechanism.
4. Is it feasible to replace the R1, R2 and R3 groups of the substrates with alkyl groups? If attempts have been made by the authors with available information, it is recommended that they briefly mention these attempts in the main text or supplementary information, because the readers might be interested in it even if such substrates cannot work for this reaction.
5. In the mechanistic section, the authors propose that the interaction between calcium phosphate and the carbonyl group of o-QMs controls the enantioselectivity of the reaction. It is recommended that the authors conduct experiments when the phenolic hydroxyl group of the o-hydroxybenzyl alcohols is protected. This will provide stronger evidence for the mechanism.
6. In Figure 4, the label below the o-hydroxybenzyl alcohols should be '1' instead of '1a'. Additionally, the title "Substrates scope of o-QMs" is inappropriate, as it also contains 2-(1-alkynyl)-2-alken-1-ones with various substituents.
7. To further demonstrate the practicality of this method, the scale-up reaction may be conducted.

Reviewer 1:

This work realizes a relay and enantioselective reaction using combined chiral gold and chiral calcium phosphate catalysts. The relay catalysis is sure to be unprecedented although the key intermediates in the first step of gold catalysis is well established. The second step using a 1:1 reactant ratio, showing a dynamical kinetic resolution. Furthermore, there are DFT calculations to show the energy profile of the proposed reaction pathway. Overall speaking, this work is novel because Ca(+2) is very crucial as the acid (H+1) and Na(+1) gave poor enantioselectivity or poor diastereoselectivity.

A combined relay catalysis to yield furanyl ethers through a carbocation intermediate is also challenging although some reports of chiral phosphate are known to be effective albeit with limited cases. This reviewer is enthusiastic to recommend this work in this journal after minor revision.

Q1: In Figure 1, the proposed catalytic circle shows the ligands L* and A* but the left hand has the ligands L₃ and L₆; these ligands are actually the same and should be clearly labelled by L₃ and L₆ in the catalytic circle.

A1: Thank you for your suggestion. We deleted the L* and A* and replaced them with L₃ or L₆.

Q2: In Figure 2, entries 9 and 10 are testing two enantiomers of L₃ ligands; but the outcome of ee are nearly the same. But in entries 16 and 17, the resulting enantiomers are reverse. Why there is so different for the outcome of enantioselectivity.

A2: Actually, we were also confused by these results at first. However, upon conducting Density Functional Theory (DFT) calculations, a comprehensive elucidation of the outcomes became apparent. In the catalytic system comprising AuL₃ and S-CaA* (Fig. 8), the energy barrier (ΔG_2) between S-L₃-TS2B-R and S-L₃-TS1 stands at 10.2 kcal/mol, surpassing the energy barriers (ΔG_1 or $\Delta G_1'$) between S-L₃-TS1 and S-L₃-1, or R-L₃-TS1 and R-L₃-1. This indicates that the rate-limiting step in the catalytic process is the Ca catalysis, and the absolute configuration of the product is contingent upon the Ca catalyst. Consequently, entries 9 and 10 pertain to testing two enantiomers of L₃ ligands, yet the outcomes in terms of enantiomeric excess (ee) are nearly identical.

In the catalytic system involving AuL₆ and *S*-CaA* (Fig. 9), the energy barrier (ΔG_1) between *R*-L₆-TS1 and *R*-L₆-1 is recorded at 8.4 kcal/mol, exceeding the energy barriers (ΔG_2 , ΔG_3 , or $\Delta G_3'$). This signifies that the initial Au catalysis step possesses the highest activation energy and serves as the rate-limiting step. The absolute configuration of the product is contingent upon the Au catalyst. Consequently, entries 16 and 17 pertain to testing two enantiomers of L₆ ligands, resulting in reversed enantiomer

Q3: In Figure 2, N.D. is not clear to me. Is no reaction taking place or no enantioselectivity or no diastereoselectivity.

A3: We added the explanation in figure 2. N.D. means no detection. There is no reaction in entries 3, 4 or 5. Therefore, we do not test the ee value. In addition, the d.r. value of product in entries 14 or 15 reactions is 1:1. There is no major product and we do not give the ee value of major product.

Q4: In the Data availability, there are x-ray crystallographic work to be deposited at Cambridge Crystallographic Data center. For the convenience of readers; ORTEP images of **3a** and **4j** (line 182) must be provided in the text as I almost reject this paper when I did not see these very important data in the text.

Q5: There two x-ray structures mentioned in the text, but only CCDC 2125710 is mentioned line 319?

A4, A5: Thank you for your suggestion! We could not obtain the single crystal of compound **3** or **4** after many experiments, because they are foam solid. Fortunately, the single crystal of derivative product **6b** from **3w** (catalyzed by L₃) were achieved. Following your suggestion, we provide the ORTEP images of compound **6b** in the text (figure 5). However, we could not get the single crystal of derivative **6** from **4** (catalyzed by L₆). To obtain the absolute configuration of product **4**, we conducted the CD experiments and theory computational spectra which have been proven to be available methods for determining the relative configuration and absolute configuration (Ref 81-83). The details were listed in the supporting information.

Q6: DFT Figures in Figure 8 and 9 are very small to show the clarity. I am unable to make a suggestion and probably a vertical presentation is better.

A6: Thank you for the helpful suggestion. We let these figures vertical and made it more clear.

Reviewer 2:

A Ca-/Au-cocatalyzed, enantio- and diastereoselective reaction of α -alkynyl- α,β -unsaturated ketones with ortho-hydroxy benzyl alcohols via transient ortho-quinone methides, 3-furyl methyl cations, and in situ generated water is reported giving rise to hindered chiral tetra-aryl substituted ethers. Moderate to good yields and good to very good levels of diastereoselectivities and enantioselectivities were reported. Depending upon the proper choice of catalyst combination all four enantio- and diastereomers were available at will. Mechanistically, the water generated in situ from dehydration of the ortho-hydroxy benzyl alcohol attacks both the 3-furyl methyl Au(I) complex and the ortho-quinone methide. Altogether, a nice combination of chiral Ca(II)/Au(I) bimetallic catalysis controlling two stereogenic centers via a cascade reaction in a single operation. DFT-calculations corroborate the mechanistic picture outlined above.

While I find the work has been carried with great care and reports excellent selectivities, dual or cooperative catalysis has been established quite a number of times with organo/metal phosphate catalysis limiting the originality of the work. In addition, the products are rather special in their structure and I am wondering what they can be employed for. The ensuing synthetic transformation reported in Figure 5 does not convince the reader of the synthetic value of this transformation.

A: Thank you for your suggestion. The organo/metal phosphate cooperative catalysis was established, but the synergistic catalysis of chiral calcium(II) phosphate and chiral gold(I) complex is not developed. Moreover, the d.r. value in most reported synergistic catalysis depends on the absolute configuration of two chiral catalysis. However, the d.r. value in this reaction was controlled by the ligand of Au-complex.

According to the pioneer work from Phil S. Baran [*Nature* **573**, 398-402 (2019)], hindered ether, which is a key intermediate in the synthesis of an aurora kinase modulator, exemplifies this commonly faced challenge. Gregory C. Fu's work [*Nature* **618**, 301-307 (2023)] also emphasized

the development of methods for hindered carbon–oxygen bond construction with simultaneous control of stereoselectivity is an important objective in synthesis. Therefore, we consider that the hinder ether is useful in medicine and chemical engineer.

In order to evaluate the synthetic potential of this protocol, we tried to conduct synthetic transformations to build more hindered ethers with different structure. Besides the reported transformation, the hydroxyl group of the **4j**'s major diastereomer was protected with propargyl. We further performed a Cu-catalyzed cycloaddition to synthesize **7** with a triazole skeleton in 76% yield and 93% ee according to “click chemistry”. **4a** could be easily triflated for subsequent efficient cross-coupling to form biaryl **8**, with the benzylic stereocenter remaining intact. Notably, no erosion in enantiopurity was observed in these transformations. We added these two synthetic transformations in the manuscript (Figure 5c).

Reviewer #3 (Remarks to the Author):

Chiral hindered ethers are ubiquitous motifs in natural and bioactive products. In this manuscript, Li and co-workers reported a method of synthesizing chiral hindered ethers by Calcium(II)/Gold(I) synergistic catalysis. The combined catalyst system features the use of chiral Ca(II) phosphate for the generation and activation of o-QMs, while chiral Au(I) catalyst is used for the generating and activating 3-furyl methyl cations. These two chiral catalysts allow for full control over the configuration of the stereocenters, and four stereoisomers of the tetra-aryl substituted ethers can be obtained by selecting suitable chiral ligands. Despite these advantages, however, the completeness and richness of this work may not be sufficient for publication in Nature Communications.

Comments:

Q1. The structure of the product is inconsistent with that of the reactants in Fig 1a, and in Fig 1d, substrate **1** generates Int-2 under the action of Catalyst CaA*, hence CaA* should be placed on the right side of Fig 1d, while Ligands **L₃** and **L₆** should be positioned on the left side for the clarity.

A1: Thank you for your suggestion. We revised the structure of the product in Fig 1a. In addition, **CaA*** was placed on the right side of Fig 1d and **L₃** and **L₆** were positioned on the left

side

Q2. In Fig 6, the authors mentioned compounds **3b'** and **4a'**, however, their detailed information was not found in the SI.

A2: Thank you for your suggestion. We added the detailed information of **3b'** and **4a'** in the SI.

Q3. The authors propose that under Au(I) catalysis, 2-(1-alkynyl)-2-alken-1-ones undergo transformation into a positively charged five-membered ring, subsequently forming alcohols when reacting with water. The isolation of the alcohols may be tried to further substantiate this catalytic mechanism.

A3: Thank you for your suggestion. Following your suggestion, we tried to isolate the intermediate alcohol but failed. It means that alcohol is a high active intermediate in this reaction system.

Q4. Is it feasible to replace the R¹, R² and R³ groups of the substrates with alkyl groups? If attempts have been made by the authors with available information, it is recommended that they briefly mention these attempts in the main text or supplementary information, because the readers might be interested in it even if such substrates cannot work for this reaction.

A4: Thank you for your suggestion. We replaced R¹ with cyclohexyl. Pleasingly, alkyl-substituted *o*-QM was shown to be suitable acceptors for the reaction, giving rise to the corresponding product **3o** in moderate yield with 74% ee by using (*R*)-L₃[(NCMe)AuSbF₆] as Au complex. However, when (*R*)-L₆[(NCMe)AuSbF₆]₂ was replaced as Au complex, no product was obtained even in room temperature.

Next, we replaced R² with *n*-butyl. Pleasingly, alkyl-substituted 2-(1-alkynyl)-2-alken-1-one was well tolerated for the reaction, giving rise to the corresponding product **4m** with 86% ee by using (*R*)-L₆[(NCMe)AuSbF₆]₂ as Au complex. However, when (*R*)-L₃[(NCMe)AuSbF₆] was replaced as Au complex, no product was obtained even in room temperature.

In addition, we replaced R³ with methyl. However, no product was obtained in these reactions with (*R*)-L₃[(NCMe)AuSbF₆] or (*R*)-L₆[(NCMe)AuSbF₆]₂ as Au complex even in room

temperature probably due to the low activity of 2-(1-alkynyl)-2-alken-1-one.

The above results were added to the manuscript.

Q5. In the mechanistic section, the authors propose that the interaction between calcium phosphate and the carbonyl group of *o*-QMs controls the enantioselectivity of the reaction. It is recommended that the authors conduct experiments when the phenolic hydroxyl group of the *o*-hydroxybenzyl alcohols is protected. This will provide stronger evidence for the mechanism.

A5: Following your suggestion, we synthesis protected *o*-hydroxybenzyl alcohol **10** for the controlled experiments. No products were obtained in the reactions between **10** and **2a**. We added these control experiments in the manuscript (Figure 7d).

Q6. In Figure 4, the label below the *o*-hydroxybenzyl alcohols should be '1' instead of '1a'. Additionally, the title “Substrates scope of *o*-QMs” is inappropriate, as it also contains 2-(1-alkynyl)-2-alken-1-ones with various substituents.

Q6: Thank you for your suggestion. We changed **1a** to **1**. Moreover, we changed “the substrates scope of *o*-QMs” to “2-(1-alkynyl)-2-alken-1-ones and *o*-QMs”

A7. To further demonstrate the practicality of this method, the scale-up reaction may be conducted.

Q7: Thank you for your suggestion. The cascade reaction of **1b** and **2a** was carried out on a gram scale under optimized reaction conditions. The corresponding product, the hindered ether **4a**, could be obtained with 52% yield, 4:1 d.r. and 90% ee. **3a** was obtained with 42% yield, 8:1 d.r. and 90% ee. We added this result in the manuscript (Figure 5a).

REVIEWERS' COMMENTS

Reviewer #1 (Remarks to the Author):

In my last review, I am positive with this work and also raise concern with six scientific questions. The Authors have carefully addressed with suitable revisions. I am now very satisfied and the current format will be acceptable to me.

Reviewer #3 (Remarks to the Author):

In this manuscript, the authors reported a method for diastereo- and enantioselective synthesis of hindered ethers from α -alkynyl- α,β -unsaturated ketones and ortho-hydroxy benzyl alcohols through Calcium(II)/Gold(I) synergistic catalysis. In addition, using the same starting materials, four stereoisomers of tetra-aryl substituted ethers can be obtained by adjusting the configuration of the two catalysts.

I have thoroughly examined the initial submission of this manuscript and am pleased to report that the authors have diligently revised it, taking into account both my comments and suggestions, as well as those provided by other reviewers. The revised version now meets the standards for publication in Nature Communications and is ready for publication.